# Occurrence of Carbapenemases, Extended-Spectrum Beta-Lactamases and AmpCs among Beta-Lactamase-Producing Gram-Negative Bacteria from Clinical Sources in Accra, Ghana

**DOI:** 10.3390/antibiotics12061016

**Published:** 2023-06-05

**Authors:** Felicia A. Owusu, Noah Obeng-Nkrumah, Esther Gyinae, Sarkodie Kodom, Rhodalyn Tagoe, Blessing Kofi Adu Tabi, Nicholas T. K. D. Dayie, Japheth A. Opintan, Beverly Egyir

**Affiliations:** 1Department of Bacteriology, Noguchi Memorial Institute for Medical Research, University of Ghana, Accra 00233, Ghana; owusufelicia90@gmail.com (F.A.O.); rhodalyntagoe1310@gmail.com (R.T.); kofitabi39@gmail.com (B.K.A.T.); 2Department of Medical Laboratory Sciences, School of Biomedical and Allied Health Sciences, University of Ghana, Accra 00233, Ghana; noahobengnkrumah@gmail.com; 3Department of Microbiology, Korle-Bu Teaching Hospital, Accra 00233, Ghana; 4University of Ghana Hospital, Accra 00233, Ghana; sarkodiekodom@yahoo.com; 5Department of Medical Microbiology, University of Ghana Medical School, University of Ghana, Accra 00233, Ghana

**Keywords:** beta-lactamases, AmpC, extended-spectrum beta-lactamase, carbapenemases, third-generation cephalosporin resistance, Ghana, Africa

## Abstract

Beta-lactamase (β-lactamase)-producing Gram-negative bacteria (GNB) are of public health concern due to their resistance to routine antimicrobials. We investigated the antimicrobial resistance and occurrence of carbapenemases, extended-spectrum β-lactamases (ESBLs) and AmpCs among GNB from clinical sources. GNB were identified using matrix-assisted laser desorption/ionization time of flight–mass spectrometry (MALDITOF-MS). Antimicrobial susceptibility testing was performed via Kirby–Bauer disk diffusion and a microscan autoSCAN system. β-lactamase genes were determined via multiplex polymerase chain reactions. Of the 181 archived GNB analyzed, *Escherichia coli* and *Klebsiella pneumoniae* constituted 46% (n = 83) and 17% (n = 30), respectively. Resistance to ampicillin (51%), third-generation cephalosporins (21%), and ertapenem (21%) was observed among the isolates, with 44% being multi-drug resistant (MDR). β-lactamase genes such as AmpCs ((*bla_FOX-M_* (64%) and *bla_DHA-M_* and *bla_EDC-M_* (27%)), ESBLs ((*bla_CTX-M_* (81%), other β-lactamase genes *bla_TEM_* (73%) and *bla_SHV_* (27%)) and carbapenemase ((*bla_OXA-_*_48_ (60%) and *bla_NDM_* and *bla_KPC_* (40%)) were also detected. One *K. pneumoniae* co-harbored AmpC (*bla_FOX-M_* and *bla_EBC-M_*) and carbapenemase (*bla_KPC_* and *bla_OXA-_*_48_) genes. *bla_OXA-_*_48_ gene was detected in one carbapenem-resistant *Acinetobacter baumannii.* Overall, isolates were resistant to a wide range of antimicrobials including last-line treatment options. This underpins the need for continuous surveillance for effective management of infections caused by these pathogens in our settings.

## 1. Introduction

Gram-negative bacteria (GNB) are widespread in nature and cause life-threatening infections in humans [1]. The expression of beta-lactamase (β-lactamase) enzymes such as class C cephalosporinases (AmpCs), extended-spectrum β-lactamase (ESBL) and carbapenemase in GNB has been linked to increased antibiotic resistance resulting in therapeutic failure, prolonged hospital stays, increased healthcare cost and mortality [2].

Antimicrobial resistance (AMR) observed among GNB due to the expression of β-lactamase is a global public health concern [3]. In Ghana and other parts of the world, there is over-dependence on cephalosporin beta-lactam antibiotics for the treatment of infections due to their low toxicity and high potency, which have contributed to the phenomenon of resistance [4]. β-lactamases act by disrupting the active amide bond of the beta-lactam (β-lactam) ring and can be chromosomally encoded or acquired via mobile genetic elements (MGEs) such as plasmids, integrons, and transposons among bacteria species. The MGEs serve as vectors in transmitting β-lactamase (*bla*) and non-betalactam AMR genes among bacteria species, resulting in limited therapeutic options for the treatment of infections [5].

Plasmid-mediated AmpC β-lactamases (including *bla_FOX-M_, bla_DHA-M_, bla_CMY-M_* and *bla_ACC-M_*) are clinically relevant cephalosporinases that mediate resistance to cephalothin, cefazolin, cefoxitin, most penicillins and β-lactamase inhibitor–beta-lactam combinations. The hyperproduction of AmpC confers resistance to extended-spectrum cephalosporins including cefotaxime, ceftazidime and ceftriaxone [6]. AmpC production has been frequently reported in Enterobacterales [7]. Of concern is the hyperproduction of AmpC β-lactamases and the upregulation of efflux pumps, which may synergistically confer additional resistance to carbapenems in GNB [8]. Most importantly, AmpC-producing bacteria are a cause of healthcare-associated infections leading to strained therapeutic options [9].

The ESBL group of the β-lactamases mediates a wide range of resistance activities against frequently administered antibiotics such as penicillins, oxyimino-cephalosporins and aztreonam, but they are inactive against beta-lactam inhibitors, cephamycins (cefoxitin) and carbapenems [10]. They are predominant among Enterobacterales strains and have been associated with multi-drug-resistant (MDR) phenotypes responsible for severe and fatal infection outcomes as well as hospital and community-linked outbreaks [11]. ESBL-producing Enterobacterales are part of the World Health Organization critical priority pathogens for the research and development of novel antibiotics [12]. The most disseminated genotypes of ESBL are *bla_TEM_, bla_SHV_* and *bla_CTX-M_* [13], which are mostly borne on plasmids, and these plasmids often carry genes that mediate resistance to other classes of antibiotics such as aminoglycosides, fluoroquinolones and trimethoprim-sulfamethoxazole, further reducing options for the treatment of ESBL-associated infections [14].

Carbapenems are a drug of choice for the treatment of infections caused by ESBL- and AmpC-producing pathogens [15]. However, the emergence of carbapenemase enzymes [16] has further reduced the treatment options available for infections caused by these pathogens [17]. These carbapenemase enzymes have been associated with clinically relevant pathogens of the Enterobacterales family, *Pseudomonas aeruginosa* (*P. aeruginosa*) and *Acinetobacter baumannii* (*A. baumannii*) strains [18].

Many variants of carbapenemases have been detected, with the most predominant being KPC (class A carbapenemases), NDM and VIM (class B metallo-beta-lactamases) and the OXA-48-type (class D carbapenemases) enzymes [19]. They have been reported globally as significant contributors to healthcare-associated infections [20]. The class A enzymes efficiently resist carbapenems and inhibit the actions of clavulanic acid, albeit poorly, whilst the class B enzymes inhibit the activities of β-lactamase inhibitors and carbapenems but not aztreonam [21]. On the other hand, class D carbapenemases, which are mostly isolated from *A. baumannii, P. aeruginosa* and less commonly in the Enterobacterales family, partially inhibit clavulanates and weakly hydrolyze carbapenems [22].

In Ghana, although GNB-harboring β-lactamases are established as widespread, most studies have only investigated the presence of ESBLs and carbapenemases in Enterobacterales with emphasis on *Escherichia coli* (*E. coli*) and *Klebsiella pneumoniae* (*K. pneumoniae*) [4,23]. There is therefore a paucity of studies that have reported the concurrent presence of β-lactamases such as AmpC, ESBL and carbapenemases in an array of GNB from clinical sources. Data on the antimicrobial resistance of β-lactamase-producing organisms are essential in guiding antimicrobial therapy, effective surveillance and infection prevention and control. This study therefore investigated the occurrence and antimicrobial resistance of AmpCs-, ESBLs- and carbapenemases among a collection of GNB recovered from clinical sources.

## 2. Results

### 2.1. Spectrum of Gram-Negative Bacteria

Out of the 181 GNB processed, 89% (n = 161/181) were Enterobacterales, comprising *E. coli* (n = 83/161; 52%), *K. pneumoniae* (n = 30/161; 19%), *Proteus mirabilis* (n = 18/161; 11%), *Enterobacter* spp. (n = 16/161; 10%), *Salmonella* spp. (n = 8/161; 5%), *Citrobacter* spp. (n = 3/161; 2%), *Providencia* spp. (n = 2/161; 1.2%) and *Klebsiella oxytoca* (n = 1/161; 0.6%). *Pseudomonas aeruginosa* (n = 8/181; 4.4%), *Acinetobacter* spp. (n = 4/181; 2.2%), *Neisseria* spp. (n = 3/181; 1.7%), *Pseudomonas stutzeri* (n = 1/181; 0.5%) and others (n = 4/181; 2.2%) were also detected. Table 1 shows bacteria species identified and clinical sources.

### 2.2. Antimicrobial Resistance Pattern

The top three antibiotics with the highest susceptibility were cefoxitin (n = 144/168; 86%), ertapenem and norfloxacin (n = 133/168; 79%). Cefuroxime and cefotaxime resistance was detected among 29% (n = 48/168) of the isolates. Ceftazidime (n = 47/168; 28%) and ampicillin (n = 86/168; 51%) resistance were also detected. For ertapenem, *E. coli* isolates recorded the highest level of resistance (n = 18/83; 22%). *Pseudomonas aeruginosa* (n = 8) and *Pseudomonas stutzeri* (n = 1) were susceptible to all antibiotics tested. Among the *Acinetobacter* spp. (n = 4), *Acinetobacter nosocomialis* (n = 1/4; 25%) showed resistance to trimethoprim/sulfamethozaxole with an MIC range >2/38 µg/mL. Ampicillin was the least effective antibiotic against all strains, with a resistance prevalence of 51%. Overall, 44%, (n = 74/168) of the isolates were multi-drug resistant (MDR). The three isolates with the highest MDR proportions were *E. coli* (n = 40/74; 54%), *K. pneumoniae* (n = 11/74; 15%) and *Enterobacter* spp. (n = 11/74; 15%). The MDR isolates were susceptible to cefoxitin (n = 52/74; 70%) and ertapenem (n = 42/74; 57%). Table 2 shows the antimicrobial resistance profiles of the isolates.

### 2.3. Phenotype and Resistance Gene Markers for AmpC, ESBL and Carbapenemase

All isolates except *P. aeruginosa (n = 8)*, *Acinetobacter* spp. (n = 4), and *P. stutzeri (n = 1)*, were analyzed for the production of AmpC and ESBL, phenotypically and via polymerase chain reaction (PCR). Twenty-eight (17%) isolates were phenotypically resistant to cefoxitin (presumptive AmpC producers) and were screened via PCR for the presence of plasmid-mediated AmpC β-lactamase genes. Eleven (39%) isolates were positive for AmpC gene variants. Thirty-six isolates (36/168; 21%) were phenotypically positive for ESBL production; of these, 29 (81%) contained *bla_CTX-M_* genes. *E. coli* were the predominant carriers of ESBL genes (n = 22/29; 76%), followed by *K. pneumoniae* 24% (n = 7/29).

Of the thirty-five ertapenem-resistant isolates, five were positive for carbapenemase genes, and these included two *K. pneumoniae*, one *E. coli*, an *A. baumannii* and a *Providencia vermicola* (*P. vermicola*). Table 3 shows the phenotypic and genotypic distribution of β-lactamase amongst the isolates.

### 2.4. Resistance Gene Distribution among Isolates

AmpC genes were found among *E. coli* (n = 4/11; 36%), *K. pneumoniae* (n = 4/11; 36%), *Salmonella* spp. (n = 2/11; 18%) and *P. vermicola* (n = 1/11; 9%) isolates. These isolates originated from urine (n = 8), blood (n = 2) and wound (n = 1) samples. The most frequently identified AmpC gene was *bla_FOX-M_* (n = 7/11; 64%), observed in four *K. pneumoniae* (36%), two *E. coli* (18%) and one *Salmonella Typhi* (9%). Other AmpC genes were *bla_EBC-M_* (n = 3/11; 27%), identified in two *K. pneumoniae*, and one *Salmonella paratyphi*. The *bla_DHA-M_* gene (n = 3/11; 27%) was found in two *E. coli* and one *P. vermicola* isolates. Two out of four *K. pneumoniae* co-harbored *bla_FOX-M_* and *bla_EBC-M_* genes. The 29 isolates harboring the ESBL- *bla_CTX-M_* gene were isolated from urine (n = 11) and wound (n = 18) samples. Other β-lactamase genes, such as *bla_TEM_* (n = 16/22; 73%) and *bla_SHV_* (n = 6/22; 27%), were also observed among the phenotypically positive ESBL producing isolates. *bla_TEM_ gene* was found in *E. coli* (n = 9/16; 56%) and *K. pneumoniae* (n = 7/16; 44%). The *bla_SHV_* gene was harbored by *K. pneumoniae* (n = 5/6; 83%) and *E. coli* (n = 1/6; 17%). In all, four (11%) isolates harbored a *bla_CTX-M,_ bla_TEM_* and *bla_SHV_* β-lactamase genes whereas eleven (31%) contained both *bla_CTX-M_* and *bla_TEM_* genes. Five isolates (14%) possessed both *bla_CTX-M_* and *bla_SHV_* genes. The *bla_OXA-_*_48_ gene was the most common carbapenemase gene identified and was carried by three of the five carbapenem-resistant GN isolates (*K. pneumoniae*, *E. coli* and *A. baumannii*). Two of the isolates with *bla_OXA-_*_48_ (*E. coli* and *K. pneumoniae)* were from urine samples and each co-harbored a *bla_KPC_* gene. *A. baumanni* carried *bla_OXA-_*_48_ and was recovered from a blood sample. *bla_NDM_* was identified in *K. pneumoniae* (n = 1) and *P. vermicola* (n = 1) isolates, both from wound samples. Figure 1 shows beta lactamase distribution and antibiotypes.

## 3. Discussion

This study found high levels of β-lactamase genes among GNB. The finding that *E. coli* and *K. pneumoniae* were the predominant GNB is similar to previous studies conducted in the country [4] as well as other parts of the world [3]. These organisms have been associated with antibiotic resistance and implicated in series of infections associated with high morbidity and mortality rates [24].

The most effective antibiotic in the study was cefoxitin. Cefoxitin, a cephamycin/second-generation cephalosporin, is known to exhibit bactericidal actions against GNB and is very potent in vitro against ESBL producers [25]. All the ESBL-producing GNB in our study, except one, were susceptible to cefoxitin. It is known to be a suitable alternative for carbapenems in infection treatment [25]. A substantial proportion of the study isolates were susceptible to fluoroquinolone, contrary to other studies in the country [26] and elsewhere (Nigeria, Tanzania and Rwanda) [27,28,29]. Our findings could be an indication that such a class of antibiotic might still be relevant in the treatment of GNB-associated infections in Ghana.

The proportion (22%) of ESBL phenotypes observed among the Enterobacterales is in line with studies in the country [30] and in other parts of the world [31,32], with *E. coli* and *K. pneumoniae* as frequent ESBL producers. The most predominant ESBL gene was *bla_CTX-M_* (81%), which is the case globally [33,34,35]. On the contrary, *bla_TEM_* was the predominant gene in a recent study in Ghana [30] and Nigeria [36]. The presence of *bla_CTX-M_* represents a significant public health threat since isolates harboring this gene co-harbor extra antimicrobial resistance genes making them resistant to a wide range of antimicrobials [37]. This is particularly worrying since these drugs are part of treatment regimens for the management of infections in Ghana [8]. The co-harboring of *bla_CTX-M_* and other genes could foster the growth and dissemination pathogens with multiple resistance genes in the country [37].

The finding of greater proportions of AmpC-producing GNB being MDRs was expected since the presence of AmpC β-lactamases is accompanied with resistance to first to third-generation cephalosporins, including cephymacins. It is known that AmpC enzymes may disguise the true effect of ESBLs and their identification in a bacteria strain, thus complicating the treatment of infections caused by bacteria co-harboring both genes [38]. In this study, *E. coli* co-harbored AmpC and ESBL (*bla_TEM_* and *bla_DHA-M_*) genes; likewise, *K. pneumoniae* co-harbored AmpC and carbapenemase genes ((*bla_FOX-M_* + *bla_EBC-M_*) + (*bla_KPC_ + bla_OXA-_*_48_*)*), which has also been observed in studies in Egypt [38] and Nigeria [39].

Carbapenems are the drug of choice for the treatment of severe infections [39]. In our study, the carbapenem-tested ertapenem was one of the most effective antibiotics, and this is an assurance that this drug class is still effective. However, the *bla_KPC_* gene, which is known to hydrolyze carbapenems and was initially identified in the north-eastern part of the US, has now spread throughout the US and most of the world [40]. Our study isolated carbapenem-resistant *K. pneumoniae* co-harboring *bla_KPC_* and *bla_OXA-_*_48_. To the best of our knowledge, this is one of two studies in the country to record a dual detectable carbapenemase gene (*bla_KPC_* and *bla_OXA-_*_48_) in a single carbapenem-resistant *K. pneumoniae* strain [41]. The presence of *bla_OXA-_*_48_ and *bla_NDM_* among GNB has been reported in other studies in Ghana [42]. Furthermore, we observed that the carbapenem-resistant *A. baumanii* (crAb) isolated from a blood culture harbored a *bla_OXA-_*_48_ gene in this study. In Ghana, Monheimer and colleagues identified crAb (77%) that harbored *bla_OXA-_*_23_, and *bla_NDM_* genes [43]. Carbapenem-resistant *A. baumannii* are among the WHO-prioritized organisms under surveillance worldwide since they are associated with life-threatening infections and the frequent carriage of multi-drug resistance [17]. Their isolation from a blood sample means that such a patient will not respond to the routinely used antimicrobials. Meanwhile, data are still limited in this part of Africa on the characteristics of this pathogen [44]. As far as we know, this is the first report of a *bla_OXA-_*_48_ crAb recovered from a blood culture in the country. In this study, there was one *K. pneumoniae* strain that co-harbored carbapenemase (*bla_KPC_* + *bla_OXA-_*_48_) and AmpC genes (*bla_FOX-M_* + *bla_EBC-M_*). Kaizeman in Iran had a similar result, but his *K. pneumoniae* isolates harbored ESBL genes (*bla_CTX-M_*, *bla_TEM_* and *bla_SHV_*) and a *bla_VIM_* carbapenemase gene [45].

The ability to harbor multiple resistance genes by GNB gives them the advantage to thrive longer in healthy individuals or the environment or cause infections in immune-compromised victims. The resistant genes identified in this study fail infection control systems and subsquently lead to dire outcomes [17]. There is the need to improve infection control measures, strengthen antimicrobial resistance surveillance and identify new treatment strategies in order to mitigate this problem. 

## 4. Materials and Methods

### 4.1. Identification of Bacteria

This study analyzed 181 archived GNB recovered from clinical sources from April 2017 to April 2018 as a form of laboratory surveillance of antimicrobial resistance and β-lactamase production. Isolates originated from urine (n = 114), wounds (n = 38), blood (n = 21), throat swabs (n = 4), stools (n = 3), and an ear swab from humans (outpatients and inpatients) with diverse infections. The isolates were sub-cultured on blood and MacConkey agar. The identification and confirmation of bacteria species was achieved using matrix-assisted laser desorption/ionization time of flight (MALDI-TOF)–mass spectrometry (Bruker Daltonics, Bremen, Germany) using Biotyper™ system 2.0 software at the genus (log(score) 1.7–2.0)) and species (log(score) ≥ 2.0) level.

### 4.2. Antimicrobial Susceptibility Testing

Kirby–Bauer disk diffusion was performed using the 0.5 McFarland standard of bacteria suspension seeded into Mueller–Hinton agar plates. The antibiotics tested included ampicillin (10 µg), cefotaxime (30 µg), norfloxacin (10 µg), ceftazidime (30 µg), ertapenem (10 µg), cefoxitin (30 µg) and cefuroxime (30 µg). The measured zone sizes were interpreted according to the Clinical and Laboratory Standard (CLSI, M100, 26th ed, 2018) guidelines (CLSI 2018). *P. aeruginosa* was tested against gentamicin (10 µg), cefepime (30 µg), ceftazidime (30 µg), ciprofloxacin (5 µg), piperacillin tazobactam (110 µg), amikacin (30 µg) and meropenem (10 µg). *Pseudomonas stutzeri* and *Acinetobacter* spp. were tested against 11 antibiotics in a microbroth dilution using the MicroScan autoSCAN-4-System with the NC 66 panel (Beckman Coulter Life Sciences, Indianapolis, IN, USA) following the manufacturer’s instructions. *K. pneumoniae* ATCC 700603 and *E. coli* ATCC 25922 were used as control strains in the evaluation of the performance of the tests. An MDR phenotype was defined as non-susceptibility to ≥1 agent in ≥3 antimicrobial categories [46]. Isolates with non-susceptibility to 3rd-generation cephalosporins were examined for the possible presence of AmpCs and ESBLs according to the CLSI guideline.

### 4.3. Phenotypic Screening for AmpC, ESBL and Carbapenem Resistance

Investigations for AmpC, ESBL and carbapenemases were only carried out for Enterobacterales. Other GNB were screened for carbapenemase production. Using the disk diffusion assay, the presumptive detection of AmpC was performed by observing reduced susceptibility (inhibition zone < 18 mm) to cefoxitin (30 µg). The expression of ESBLs was ascertained via the combined double-disk method using cefotaxime (30 µg) and ceftazidime (30 µg) alone and in combination with clavulanic acid (10 µg). An inhibition zone difference of ≥5 mm between the single and the clavulanic acid combination disks for cefotaxime and ceftazidime confirmed ESBL expression. Isolates resistant to ertapenem (10 µg) (inhibition zone ≤ 15 mm) were deemed possible carbapenemase producers. These isolates were subjected to a confirmatory test using the Modified Hodges Test (MHT) and the modified Carbapenem Inactivation Method (mCIM) according to the CLSI guidelines. *K. pneumoniae* ATCC BAA 1705 and *E. coli* ATCC 25922 strains were used as positive and negative controls, respectively.

### 4.4. Molecular Detection of Antimicrobial-Resistant Gene Markers of Beta-Lactamases

Isolates that tested positive for AmpC, ESBL, or carbapenemase phenotypes were subjected to PCR to confirm the genes encoding for AmpC (*bla_MOX-M_, bla_ACC-M_, bla_EBC-M_, bla_FOX-M_, bla_CIT-M_ and bla_DHA-M_*) [47], ESBLs (*bla_CTX-M_*) [48], carbapenemases (*bla_KPC_, bla_NDM_, bla_VIM_, bla_IMP_ and bla_OXA-_*_48_) [49], and other β-lactamase genes (*bla_TEM_* and *bla_SHV_*) [48] with slight modifications. Crude DNA (10 µL) was extracted from pure overnight cultures and suspended in 200μL of molecular-grade nuclease-free water, heated for 10 min at 98 °C, and centrifuged for 5 min at 4 °C and 20,000 g, as previously suggested by Quansah [42]. The supernatant was transferred into sterile 1.5 mL Eppendorf^®^ tubes and used as a template for the PCR. For the PCR amplification, each reaction mix of 25 µL consisted of 12.5 uL of Green PCR Master Mix (2×) (DreamTaq, Thermo Scientific, Waltham, MA, USA), 4.5 µL of primer mix, 6 µL of molecular-grade nuclease-free water and 2 µL of crude DNA template, as previously demonstrated by Khurana et al., 2018 [49]. The primers used for PCR amplification and cycling conditions are listed in Table 4. All PCR amplicons were analyzed via horizontal gel-electrophoresis in a 2% (weight/volume) agarose gel (SeaKem^®^GTG^®^Agarose, Lonza, Basel, Switzerland) using Tris/Acetate/EDTA 50× concentrate buffer.

### 4.5. Data Analysis

Data from laboratory investigations were entered into a Microsoft Excel (version 2304) database for editing and analyses. Categorical variables were summarized with frequencies and percentages.

## 5. Conclusions

We show in this study that GNB are frequently resistant to antimicrobials and harbor genes for β-lactamases such as AmpCs, ESBLs and carbapenemases which are significant determinants of antimicrobial resistance. The finding of *bla_KPC_* and *bla_OXA-_*_48_ producing GNB is worrying, especially in the absence of the routine detection of these pathogens in most of the clinical microbiology laboratories in the country. The study results further support the need for efficient AMR surveillance systems at local and national levels to monitor the rise and spread of these beta-lactamase-producers and other multi-drug resistant pathogens in the country.

## 6. Limitations

Not all the classes of AmpC, ESBLs and carbapenemase genes were screened using PCR. Thus, resistant isolates harboring other relevant resistance genes were not identified. Because the genomes of the isolates were not sequenced, the specific types of *TEM* and *SHV* β-lactamase, as well as the *OXA-48* identified in *A. baumannii*, were not determined.

## Figures and Tables

**Figure 1 antibiotics-12-01016-f001:**
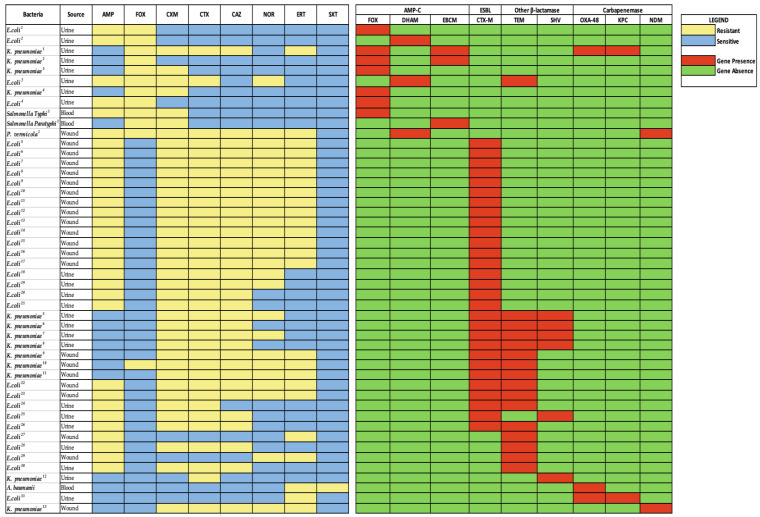
β-lactamase gene distribution and antibiotypes of isolates. Yellow cells indicate GNB that are resistant to antibiotics, whilst blue shows susceptibility to antibiotics. The red-colored cells also depict which genes the isolate harbors, and the green-colored cells depict the absence of a gene in the isolate. NOTE: ^1–31^—Non duplicate isolates. AMP—ampicillin; CTX—cefotaxime; NOR—norfloxacin; CAZ—ceftazidime; ERT—ertapenem; FOX—cefoxitin; CXM—cefuroxime, SXT—trimethoprim sulphamethozaxole.

**Table 1 antibiotics-12-01016-t001:** Gram-negative bacteria and their clinical sources.

Isolates	Urine	Wound	Blood	Throat	Stool	Ear
	n (%)	n (%)	n (%)	n (%)	n (%)	n (%)
*Escherichia coli* (n = 83)	61 (73)	18 (22)	3 (4)	_	_	1 (1.2)
*Klebsiella pneumoniae* (n = 30)	25 (83)	3 (10)	2 (7)	_	_	_
*Proteus mirabilis* (n = 18)	8 (44)	2 (11)	7 (39)	_	1 (6)	_
*Pseudomonas aeruginosa* (n = 8)	_	8 (100)	_	_	_	_
*Pseudomonas stutzeri* (n = 1)	_	_	1 (100)	_	_	_
*Enterobacter cloacae* (n = 7)	5 (71)	_	2 (29)	_	_	_
*Enterobacter asburiae* (n = 4)	4 (100)	_	_	_	_	_
*Enterobacter kobei* (n = 4)	3 (75)	1 (25)	_	_	_	_
*Enterobacter aerogenes* (n = 1)	1 (100)	_	_	_	_	_
*Salmonella paratyphi* (n = 3)	1 (33)	_	1 (33)	1 (33)	_	_
*Salmonella typhi* (n = 3)	_	_	3 (100)	_	_	_
*Salmonella enterica* (n = 2)	_	_	1 (50)	_	1 (50)	_
*Acinetobacter baumannii* (n = 2)	1 (50)	_	1 (50)	_	_	_
*Acinetobacter nosocomialis* (n = 2)	2 (100)	_	_	_	_	_
*Neisseria subflava* (n = 2)	_	_	_	2 (100)	_	_
*Klebsiella oxytoca* (n = 1)	_	1 (100)	_	_	_	_
*Providencia stuartii* (n = 1)	_	1 (100)	_	_	_	_
*Providencia vermicola* (n = 1)	_	1 (100)	_	_	_	_
*Citrobacter koseri* (n = 1)	1 (100)	_	_	_	_	_
*Citrobacter youngae* (n = 1)	_	_	_	_	1 (100)	_
*Citrobacter freundi* (n = 1)	1 (100)	_	_	_	_	_
*Neisseria meningiditis* (n = 1)	_	_	_	1 (100)	_	_
*Kerstersia gyiorum* (n = 1)	_	1 (100)	_	_	_	_
*Achromobacter xylososidans* (n = 1)	_	1 (100)	_	_	_	_
*Alcaligenes faecalis* (n = 1)	_	1 (100)	_	_	_	_
*Cupriavidus gilardii* (n = 1)	1 (100)	_	_	_	_	_
Total (181)	114 (63)	38 (21)	21 (12)	4 (2.2)	3 (1.6)	1 (0.5)

Note: n—number.

**Table 2 antibiotics-12-01016-t002:** Antimicrobial resistance profile of isolates recovered.

Bacterial Isolates	AMP N (%)	CTX N (%)	NOR N (%)	CAZ N (%)	ERT N (%)	FOX N (%)	CXM N (%)	MDR N (%)
*E. coli* (n = 83)	76 (92)	27 (33)	23 (28)	23 (28)	18 (22)	5 (6)	29 (35)	40 (48)
*K. pneumoniae* (n = 30)	^#^_	12 (40)	5 (17)	9 (30)	6 (20)	5 (17)	10 (33)	11 (37)
*P. mirabilis* (n = 18)	4 (22)	1 (6)	0	6 (33)	2 (11)	0	1 (6)	1 (6)
*Enterobacter* spp. (n = 16)	^#^_	3 (19)	2 (13)	4 (25)	3 (19)	8 (50)	3 (19)	11 (69)
*Citrobacter* spp. (n = 3)	^#^_	1 (33)	0	0	0	2 (67)	0	2 (67)
*Salmonella Typhi* (n = 3)	2 (67)	0	0	0	0	1 (33)	$_	1 (33)
*Salmonella Paratyphi* (n = 3)	0	0	0	0	0	1 (33)	$_	1 (33)
*Neisseria* spp. (n = 3)	NA	1 (33)	NA	NA	NA	0	NA	0
*Salmonella enterica.* (n = 2)	0	0	0	1 (50)	0	0	$_	0
*K. oxytoca* (n = 1)	^#^_	0	0	0	1 (100)	0	1 (100)	1 (100)
*Providencia vermicola* (n = 1)	1 (100)	1 (100)	1 (100)	1 (100)	1 (100)	1 (100)	1 (100)	1 (100)
*Providencia stuartii* (n = 1)	1 (100)	0	1 (100)	0	1 (100)	0	0	1 (100)
*Kerstersia gyiorum* (n = 1)	0	0	1 (100)	0	0	0	1 (100)	1 (100)
*Achromobacter xylososidans* (n = 1)	1 (100)	1 (100)	1 (100)	1 (100)	1 (100)	1 (100)	1 (100)	1 (100)
*Alcaligenes faecalis* (n = 1)	1 (100)	1 (100)	1 (100)	1 (100)	1 (100)	0	1 (100)	1 (100)
*Cupriavidus gilardii* (n = 1)	NA	0	0	1 (100)	1 (100)	0	NA	1 (100)
Total (n = 168)	86 (51)	48 (29)	35 (21)	47 (28)	35 (21)	24 (14)	48 (29)	74 (44)

*P. aeruginosa* (n = 8) were susceptible to gentamicin (≥15 mm), cefepime (≥18 mm), ceftazidime (≥21 mm), ciprofloxacin (≥25 mm), piperacillin tazobactam (≥21 mm), amikacin (≥17 mm) and meropenem (≥19 mm). *Acinetobacter* sp. (n = 4)—were susceptible to amikacin (MIC ≤ 16 µg/mL), ampicillin/sulbactam (MIC ≤ 8/4 µg/mL), cefepime (MIC ≤ 1, 4–8 µg/mL), cefotaxime (MIC 8 µg/mL), ceftazidime (MIC 4 µg/mL), ciprofloxacin (MIC ≤ 1 µg/mL), gentamicin (MIC ≤ 4 µg/mL), levofloxacin (MIC ≤ 2 µg/mL), meropenem (MIC ≤ 1 µg/mL), tobramycin (MIC ≤ 4 µg/mL) and trimethoprim/sulfamethozaxole (MIC < 2/38 µg/mL). *Acinetobacter nosocomialis* (n = 1) was resistant to trimethoprim/sulfamethozaxole with MIC > 2/38 µg/mL. *Pseudomonas stutzeri* (n = 1) was susceptible to cefepime (MIC ≤ 1 µg/mL), cefotaxime (MIC ≤ 1 µg/mL), ceftazidime (MIC ≤ 1 µg/mL) and piperacillin/tazobactam (MIC ≤ 16 µg/mL). Note: NA—not applicable; ^#^_—isolates were not tested because they are intrinsically resistant to ampicillin. $_—results were not reported because 1st- and 2nd-generation cephalosporins may appear active in vitro against Salmonellae but are clinically inactive. AMP—ampicillin; CTX—cefotaxime; NOR—norfloxacin; CAZ—ceftazidime; ERT—ertapenem; FOX—cefoxitin; CXM—cefuroxime; MDR—multi-drug resistance.

**Table 3 antibiotics-12-01016-t003:** Phenotypic and genotypic distribution of AmpC, ESBL and carbapenemases among isolates.

Isolate	AmpC		ESBL		Carbapenemase	
	Phenotypic (n = 28)	Genotypic (n = 11)	Phenotypic (n = 36)	Genotypic (n = 29)	Phenotypic (n = 35)	Genotypic (n = 5)
*E. coli* (n = 83)	5	4 (80%)	27	22 (81%)	18	1 (6%)
*K. pneumoniae* (n = 30)	5	4 (80%)	8	7 (88%)	6	2 (33%)
*Proteus mirabilis* (n = 18)	-	-	1	-	2	-
*Enterobacter* spp. (n = 16)	8	-	-	-	3	-
*Salmonella* spp. (n = 8)	2	2 (100%)	-	-	-	-
*Acinetobacter* spp. (n = 4)	4	-	-	-	3	1 (33%)
*Citrobacter spp*. (n = 3)	2	-	-	-	-	-
*K. oxytoca* (n = 1)	-	-	-	-	1	-
*Providencia vermicola* (n = 1)	1	1 (100%)	-	-	1	1 (100%)
*Achromobacter xylososidans* (n = 1)	1	-	-	-	-	-
*Cupriavidus gilardii* (n = 1)	-	-	-	-	1	-

*P. aeruginosa* (n = 8), *Neisseria* spp. (n = 3), *Kerstersia gyiorum* (n = 1), *Alcaligenes faecalis* (n = 1), *Providencia stuartii* (n = 1) and *Pseudomonas stutzeri* (n = 1) were negative for AmpC, ESBL and carbapenemase production.

**Table 4 antibiotics-12-01016-t004:** PCR amplification: primer sequences and cycling conditions.

Gene	Primers (5′-3′)	Size (Bp)	Cycling Conditions	References
**ESBL**				
*CTX-M*	FP: GAAGGTCATCAAGAAGGTGCG RP: GCATTGCCACGCTTTTCATAG	560	Initial denaturation at 95 °C for 5 min, followed by 30 cycles of denaturation at 95 °C for 30 s, primer annealing at 60 °C for 30 s, extension at 72 °C for 2 min and a final elongation temperature at 72 °C for 10 min.	[48]
**OTHER BETA-LACTAMASE**				
*SHV*	FP: GTCAGCGAAAAACACCTTGCC RP: GTCTTATCGGCGATAAACCAG	383		
*TEM*	FP: GAGACAATAACCCTGGTAAAT RP: AGAAGTAAGTTGGCAGCAGTG	420		

**CARBAPENEMASE**				
*KPC*	FP: ATGTCACTGTATCGCCGTC RP: AATCCCTCCGAGCGCGAG	863	Amplification was carried out at 94 °C for 3 min as the initial step for denaturation, followed by 35 cycles of denaturation at 94 °C for 30 s, annealing at 61.6 °C for 30 s and extension at 72 °C for 1 min. Final elongation was at 72 °C for 7 min.	[49]
*OXA-48*	FP: GCTTGATCGCCCTCGATT RP: GATTTGCTCCGTGGCCGAAA	281		
*IMP*	FP: GGCAGTCGCCCTAAAACAAA RP: TAGTTACTTGGCTGTGATGG	737		
*VIM*	FP: AAAGTTATGCCGCACTCACC RP: TGCAACTTCATGTTATGCCG	865		
*NDM*	FP: GGTGCATGCCCGGTGAAATC RP: ATGCTGGCCTTGGGGAACG	660		
**AMPC**				
*MOXM*	FR: GCTGCTCAAGGAGCACAGGAT RP: CACATTGACATAGGTGTGGTGC	520	Amplification was carried out at 94 °C for 15 min as the initial step for denaturation, 25 cycles of denaturation at 94 °C for 30 s, annealing at 64 °C for 90 s and extension at 72 °C for 60 s. Final elongation was at 72 °C for 10 min.	[47]
*CITM*	FR: TGGCCAGAACTGACAGGCAAA RP: TTT CTC CTG AAC GTG GCT GGC	462		
*DHAM*	FR: AACTTTCACAGGTGTGCTGGGT RP: CCGTACGCATACTGGCTTTGC	405		
*FOXM*	FP: AACATGGGGTATCAGGGAGATG RP: CAAAGCGCGTAACCGGATTGG	190		
*ACCM*	FP: AACAGCCTCAGCAGCCGGTTA RP: TTCGCCGCAATCATCCCTAGC	346		
*EBCM*	FR: TCGGTAAAGCCGATGTTGCGG RP: CTTCCACTGCGGCTGCCAGTT	302		

## Data Availability

The data used/analyzed in this study can be made available by the corresponding author on request.

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
