# Peer review of "Occurrence of Carbapenemases, Extended-Spectrum Beta-Lactamases and AmpCs among Beta-Lactamase-Producing Gram-Negative Bacteria from Clinical Sources in Accra, Ghana"

_antibiotics, 2023, doi:10.3390/antibiotics12061016_

Round 1
Reviewer 1 Report
The main question addressed by the research: Occurrence of Carbapenemases, Extended-spectrum beta-lactamases and AmpCs among Beta-lactamase Producing Gram-negative Bacteria from Clinical Sources in Accra, Ghana. The emergence and spread of antibiotic resistance Gram-negative bacteria represent a serious public health problem.
The references appropriate. The number of references (78) meet the requirements of the journal for articles in addition, the number of sources five years ago (2018-2023) is only 33,3% (26), which is insufficient.
Not all the classes of AmpC, ESBLs, and carbapenemase genes were screened using PCR, for example no information about harbouring blaOXA-23, blaOXA-51 , blaOXA-58 in A. baumannii.
In this article author’s detected blaOXA-48 in A. baumannii by PCR, but blaOXA-48 usually harbouring K. pneumoniae. And because the genomes of the isolates were not sequenced and could not have been included to NCBI, blaOXA-48 in A. baumannii cannot be identified.
Why was a limited number of strains from such different materials studied in the work? It is not information from whom the strains were isolated, with what diseases, which hospitals.
There is no conclusion of the ethical commission on the possibility of conducting this study.
This is a study of a limited number of isolates. In this regard, the work significance of the content is low.
To sum up, the work is amount of novel information derived from this study does not justify its publication.
Author Response
Please find the our responses in the file attached

Reviewer 2 Report
The manuscript entitles "Occurrence of Carbapenemases, Extended-spectrum beta-lactamases and AmpCs among Beta-lactamase Producing Gram negative Bacteria from Clinical Sources in Accra, Ghana" has impact in the field of antimicrobial resistance research. The data of this research will also help to development of background data development for Ghana as well as the world research community. However, few minor corrections will improve the manuscript.
Introduction is well written, however background data from Ghana is less focused. Previous studies related to current studies should be added and justify the requirement of the study.
If possible, graphical presentation of findings will make the manuscript more readable.
For the uniformity of the manuscript, add meaning of short form for the first time when using, no need to repeat. For example, see line no. 47 and 271. Check the uniformity for all short forms.
Please add note under each table mentioning the meaning of short form.
Make the gene names as italic in the whole manuscript.
Table 1: Please add a note mentioning of "n". Remove the extra "." dot after Escherichia. Line no. 248-249: Please mention the type of laboratory weather it is a diagnostic or research laboratory, human or animal based.
Line no. 268-269: Please rephrase the sentence.
Line no. 292: Please check the sentence.
To me discussion part is relatively weak section in this manuscript. There is no statistical analysis. The analysis would help to make relations among the findings. Then better discussion writing is possible. As the samples were from clinic, there was also opportunity to show age wise relation of AMR and based on others variables. If possible, the authors may add these points. Otherwise, these are also part of limitations and could draw recommendations for future study plan.
Author Response
Please find our responses in the file attached

Reviewer 3 Report
Felicia A. Owusu et al has shown in this research article that Gram-negative bacteria (GNB) are of public health concern due to their resistance to routine antimicrobials. This study investigated the antimicrobial resistance and occurrence of carbapenemases, extended spectrum β-lactamase (ESBL) and AmpCs, among GNB from clinical sources. Escherichia coli and Klebsiella pneumoniae constituted 46% and 17% of the 181 archived GNB analyzed, respectively, with 41% being multi-drug resistant. This underpins the need for continuous surveillance for effective management of infections caused by these pathogens.
Overall, the articles look of great interest for further improvement of the article I have few suggestions:
· In line 108 – 109 write the full genus and species name of the bacteria.
· Line 149 write in before 2 E.coli
· Line 148 -149 write the 2 and 1 number in words for more clarity.
Author Response
Please our responses in the file attached

Reviewer 4 Report
The authors in this manuscript have scientific merit for publication in Antibiotics “Occurrence of Carbapenemases, Extended-spectrum beta-lactamases and AmpCs among Beta-lactamase Producing Gram-negative Bacteria from Clinical Sources in Accra, Ghana”. The authors have pointed out firstly identified GNB from various sites and the alarming emergence of multi-drug resistance in the treatment. In addition, phenotypic screening and molecular detect several genes for AmpC, ESBL, and carbapenem resistances. Overall, the study undertaken by the authors is relevant and important for the clinical setting given that such findings assist the public health sector in revising/improving guidelines for the optimal use of antibiotics in Chana. However, the manuscript needs to be rewritten for some sentences, and the scientific names of bacterial genera and species should be printed in italics as well as the legend of Table 2 needs to update. In addition, it will be better if present Table 4 as a nice figure as well as provides a figure with a legend showing the amplicon sizes of each gene and corroborating with the ladder. All genes after bla would better if italicized and subscripts such “blaFOX-M”
Comments:
Abstract:
· Line 29: Change “Microscan” with microscan
· All genes after bla would better if italicized and subscripts such “blaFOX-M”
1. Introduction:
· Lines 54-56: Remove it or try to look at the relationship between these genes and the mobile genetic elements which would increase the quality and substance of this manuscript. “β-lactamases act by disrupting the active amide bond of the beta-lactam ring and can be chromosomally encoded or acquired via mobile genetic elements such as plasmids, integrons, and transposons among bacteria species [11-13].”
· Lines 69: change “but they are inactive” with “ but inactive “
· Lines 75: No Abbreviations on the first time and write the complete names “blaTEM, blaSHV, and blaCTX-M”
· Lines 84-86: rewrite
· Line 94: change “Class” with “class”
· Line 96: consist the font size [42]
·
2. Results
· In all MS; you have to Italicize all bacterial names and check whether bacterial species will be written “sp.” or “spp.”
When the species within a given genus cannot or does not need to be identified in a sentence, it can be replaced with “sp.” If you are referring to multiple species of the same genus, “spp.” can be used. These abbreviations should not be italicized
· Lines 110-111: Revise the sentence
· Line 112: It would be better if re-order bacterial species according to genus (Klebsiella pneumoniae and Klebsiella oxytoca)
· Lines 117-118: Revise these sentences because the data on Table is about resistance and here you mentioned susceptibility.
· Line 122: change “all bacterial strains” with “all strains”
· Lines 128-134: It would be better if you present data by another Table
· Lines 140-172: revise and re-write. It prefers consist for presenting data in the 2.3. Phenotype and gene markers for AmpC, ESBL, and carbapenemase resistance
In all this subsection you mentioned numbers and then % such as line 141: Twenty-eight (22%, 28/168) isolates. It would be better if you present data as you did in previous sections starting from % (n) such as line 115 ampicillin, 48% (n=86/181) were resistant as
· Line 178: It will be easy for the reader to understand and would be better if the authors present “Beta-lactamase genes distribution and antibiotypes of isolates” in Table 4 as a figure, similar to previously published paper in Antibiotics, Figure2 in https://www.mdpi.com/2079-6382/11/9/1246 and Figures 3 and 4 https://doi.org/10.3390/antibiotics12010169
· in addition to numbrazed all strains from 1,,2 ,3….. like E.coli1, E.coli2
· A figure with a legend showing the amplicon sizes of each gene and corroborating with the ladder need to provide
·
4. Discussion
· Lines 189-194: delete (88%, n=147/168) and (79%, n=133/186) because it was already mentioned in results
· Line 192: delete (Table 4)
· Line 195: “elsewhere” would be better if u add where the country
· Lines 202-206: The sentence is too lengthy. Concise it
· Line 217: “by other studies [66,67]” would be better if u add where
· Lines 223-225: Revise the sentences
· Line 228: delete in this study
2. Materials and Methods
· Line 258; Change “into” with “unto Mueller -Hinton”.
· Line 102; Change “Polymerase Chain Reaction (PCR)” with “polymerase chain reaction (PCR)”
· Line 292; Change “bacteria DNA” with “acteria DNA”
· Lines 297-298; in the factory/ company of DreamTaq Green PCR Master
· Lines 299: add reference [78] to after “Khurana et al., 2018”.
· Lines 307-308: Double check if you are not using SPSS, delete “Statistical Package for Social Sciences (SPSS, Version 20.0) for data editing 307 and statistical analyses”
· In addition, would be recommended to add a section on the Ethical approval community if it is applicable.
Good Luck
Author Response
Please find our responses in the file attached.

Round 2
Reviewer 1 Report
No information from whom those strains were isolated (human or animal), with what diseases, which hospitals people were hospitalized
The article describes a small number of strains, especially some species.
Good luck.
Author Response
Please find the response file attached
